# Microbial Genes for a Circular and Sustainable Bio-PET Economy

**DOI:** 10.3390/genes10050373

**Published:** 2019-05-16

**Authors:** Manuel Salvador, Umar Abdulmutalib, Jaime Gonzalez, Juhyun Kim, Alex A. Smith, Jean-Loup Faulon, Ren Wei, Wolfgang Zimmermann, Jose I. Jimenez

**Affiliations:** 1Faculty of Health and Medical Sciences, University of Surrey, Guildford GU2 7XH, UK; m.salvadordelara@surrey.ac.uk (M.S.); u.abdulmutalib@surrey.ac.uk (U.A.); j.gonzalezgutierrezdelaconch@surrey.ac.uk (J.G.); juhyun.kim@surrey.ac.uk (J.K.); alex.smith@surrey.ac.uk (A.A.S.); 2Micalis Institute, INRA, AgroParisTech, Université Paris-Saclay, 78350 Jouy-en-Josas, France; jean-loup.faulon@inra.fr; 3SYNBIOCHEM Centre, Manchester Institute of Biotechnology, University of Manchester, Manchester M1 7DN, UK; 4CNRS-UMR8030/Laboratoire iSSB, Université Paris-Saclay, 91000 Évry, France; 5Department of Microbiology and Bioprocess Technology, Institute of Biochemistry, Leipzig University, 04103 Leipzig, Germany; wei@uni-leipzig.de (R.W.); wolfgang.zimmermann@uni-leipzig.de (W.Z.)

**Keywords:** plastics, biodegradation, sustainability, upcycling, biotransformations, polyethylene terephthalate, terephthalate, ethylene glycol

## Abstract

Plastics have become an important environmental concern due to their durability and resistance to degradation. Out of all plastic materials, polyesters such as polyethylene terephthalate (PET) are amenable to biological degradation due to the action of microbial polyester hydrolases. The hydrolysis products obtained from PET can thereby be used for the synthesis of novel PET as well as become a potential carbon source for microorganisms. In addition, microorganisms and biomass can be used for the synthesis of the constituent monomers of PET from renewable sources. The combination of both biodegradation and biosynthesis would enable a completely circular bio-PET economy beyond the conventional recycling processes. Circular strategies like this could contribute to significantly decreasing the environmental impact of our dependence on this polymer. Here we review the efforts made towards turning PET into a viable feedstock for microbial transformations. We highlight current bottlenecks in degradation of the polymer and metabolism of the monomers, and we showcase fully biological or semisynthetic processes leading to the synthesis of PET from sustainable substrates.

## 1. Introduction

Thermoplastic polymers, some of which constitute the majority of the commonly known plastics, are extremely useful materials endowed with properties that make them ideal for applications such as insulation and packaging [1,2]. They are durable, water-proof and versatile materials that have become almost essential in our lives. In fact, in 2017 the contribution of plastics to the European economy reached a market size of EUR 355 billion while employing 1.5 million people [3]. Plastics are light-weight and have significantly contributed to decreasing transportation costs and extending the shelf life of food [4]. Their success as a material is only comparable to their detrimental environmental impact. The accumulation of plastic waste in the environment has become an extremely serious concern [5,6]. Plastic pollution is present in every single niche of the planet, with dramatic effects on ecosystems, especially in marine environments, affecting equally large and small fauna and flora [6,7].

Plastics possess two key features: they are barely degradable by environmental physical, chemical and especially by biological processes [8], and they have low production costs, which make their reuse not economically competitive. While these individual properties are desirable, when combined they lead to the current problem we are facing: the accumulation of recalcitrant and polymers in the environment that can degrade into microplastics with potential toxic effects [9]. The story of plastic pollution is a story of mismanagement of an otherwise valuable resource. Numerous recent studies have highlighted the poor recycling rates of plastics compared to other materials. For instance, a recent report estimates the amount of virgin plastics produced from oil of over 8 billion metric tons, out of which only 9% have been recycled [10]. This reflects a saturated traditional recycling industry and emphasises the need for novel approaches to plastic management, including the possibility of harnessing microbial activities to use plastic waste as a feedstock for biotransformations [11,12,13,14].

Out of all plastics, polyesters such as polyethylene terephthalate (PET) are in a good position for becoming a sustainable polymer compared to other oil-derived counterparts. PET is obtained from the polymerisation of the constituent monomers terephthalic acid (TPA) and ethylene glycol (EG) (Figure 1). It is durable, relatively easy to mould by blowing, which results in an almost inert, hard and stiff polymer that has been adopted by the beverage industry as the main material for the production of bottles [15,16]. PET has, in addition, the highest collection rates of all plastics even though reused PET is only a small fraction of the total PET consumed: The US National Association for PET Container Resources (NAPCOR) reported that out of the approximately 3 million tons of new PET bottles reaching the market in 2017, only 29% of them were made from collected and recycled PET, a nearly 5% decrease compared to recycling rates of the previous year [17].

As a polyester, PET can be depolymerized as a more effective alternative to mechanical recycling [18]. Methods of depolymerization include glycolysis, methanolysis, hydrolysis, aminolysis and ammonolysis [19]. Among them, glycolysis has recently emerged as a key technology for recycling PET waste. Glycolysis is the process of PET degradation by glycols at high temperatures and in the presence of catalysts such as metal acetates [20]. Compared to other methods, glycolysis has the great advantage of enabling the recycling of coloured and opaque PET that cannot be otherwise recycled due to the presence of the pigments. The resulting monomers TPA and EG can be re-used to produce PET, as well as other polymers of interest [21]. Glycolysis and related methods contribute to a more sustainable PET economy, although they also have drawbacks such as the energy cost of the high temperatures required and the long reaction times needed for effective depolymerization [21].

Biological activities capable of catalysing PET hydrolysis under mild reaction conditions are emerging as an alternative to chemical PET depolymerization methods [22]. As a result, a number of enzymes from different microorganisms have been characterised [23,24,25], facilitating the implementation of PET as a biotechnological feedstock [11,26]. We argue that this strategy is more versatile than chemical methods because, if funnelled to the central microbial metabolism, the monomers obtained can be transformed into a plethora of molecules by harnessing advances in synthetic biology and metabolic engineering. This would contribute to creating a path for revenue from PET waste beyond current recycling activities. It could thereby help to mitigate the impact of PET environmental release and promote the competitive development of a next generation of environmentally friendly materials.

Given the interesting physicochemical properties of PET and its potential use as a substrate in biotechnology, in this article we review the genes that are required for a sustainable and circular PET economy. In our view, to accomplish this goal it is required to (i) improve the kinetics of PET enzymatic hydrolysis; (ii) link the metabolism of the resulting monomers to relevant biosynthetic pathways and (iii) engineer biological systems for the production of PET monomers TPA and EG from renewable sources.

## 2. PET Metabolism

The enzymatic hydrolysis of PET involves the release of constituent monomers TPA and EG due to the action of esterases. The resulting monomers can be degraded by microorganisms endowed with the appropriate metabolic pathways for these compounds. TPA is converted into protocatechuate (PCA) that will undergo dioxygenolytic cleavage and degradation through different routes prior to reaching the central metabolism [27,28,29,30]. Similarly, EG is assimilated through different pathways depending on the microorganism. For instance, it can be transformed into acetate via acetyl-CoA in *Acetobacterium woodii* [31], whereas in some strains in *Pseudomonas putida* it is funnelled directly to the Krebs cycle via isocitrate [32]. In this section we will focus on the genes responsible for these activities and their (co)occurrence in different bacterial taxa.

### 2.1. Enzymatic Hydrolysis of PET

Different types of hydrolases have shown to be active against the PET polymer. These enzymes are lipases, esterases, cutinases and carboxylesterases isolated from fungi and bacteria (see [14,25] for recent reviews on this topic). They belong to the α/β hydrolase superfamily and have evolved in a different context and for a different function [33]. For instance, the original role of the cutinases from the genus *Thermobifida* is to hydrolyse the plant polyester cutin. Among the different variants of these enzymes, the ones endowed with certain properties (e.g., a more accessible active site) display the highest activity against PET [34]. A recent bioinformatic analysis has investigated the distribution of genes encoding for homologs of these esterases in terrestrial and marine metagenomes and has allowed to identify 504 new hydrolases [35]. The two main conclusions of this study are: (i) genes potentially encoding polyester hydrolases are rare, and (ii) their taxonomic distribution seems to be related to the niche studied, with *Actinobacteria* or *Proteobacteria* being more prominent hosts in terrestrial environments, whereas *Bacteroidetes* are the most frequent hosts in marine metagenomes [35].

As a new-to-nature polymer, PET constitutes a challenge for any of the hydrolases that are active against it. In this sense, it is worth highlighting that not all types of PET are equally susceptible to microbial degradation. Depending on processing and thermal treatments, PET can occur in an amorphous form or a semi-crystalline form [36]. It has been shown that the extent of enzymatic polyester hydrolysis depends on the degree of its crystallinity and chain orientation [37]. In the amorphous regions, the polymer chains are less densely packed and are more susceptible to hydrolytic attack compared to the crystalline regions. The enzymatic degradation rate of the polyester correlates with the temperature difference between the melting temperature of the polymer and the hydrolysis temperature. The polymer chain can be considered to be more mobile and accessible to enzymatic attack when close to the glass transition temperature (Tg) of amorphous PET [38]. Therefore, increased enzymatic hydrolysis rates of PET are expected when performing the reaction at temperatures near the Tg of the amorphous polyester (67–71 °C). This suggests that efficient PET hydrolysis needs to be conducted by thermostable polyester hydrolases such as the cutinases TfCut2 and HiC isolated, respectively, from the thermophilic actinomycete *Thermobifida fusca* [23] or the fungus *Thermomyces insolens* [37], both of which, especially the latter, have been reported to be active for long periods of time at temperatures of up to 70 °C. Engineered post-translational modifications (e.g., glycosylation) can then be used on these polyester hydrolases to improve thermal properties of the enzymes further [39]. Hydrolysis at those temperatures is obviously not compatible with most bioprocesses using whole-cell catalysts, especially those involving engineered mesophilic organisms such as *Escherichia coli* which can grow up to a maximum temperature of 48–50 °C only after evolutionary adaptation and at a fitness cost [40,41]. The bacterium *Ideonella sakaiensis* has been reported to be capable of growing on PET as a sole carbon source due to the secretion of a PET hydrolase [24]. When tested in vitro and in mesophilic conditions (below the Tg of PET), this enzyme shows very low degradation rates of PET and, even though this activity could be increased somewhat by directed evolution [42], potential hydrolysis yields are far from being able to sustain industrial bioprocesses.

Another important factor affecting the performance of the enzymes hydrolysing PET is their inhibition mediated by mono-(2-hydroxyethyl) terephthalate (MHET) and bis-(2-hydroxyethyl) terephthalate (BHET), by-products of an incomplete hydrolysis [43]. These molecules are oligomers of TPA and EG that act as competitive inhibitors of the enzymes [44]. Even though it is possible to design reactors that allow for a continuous removal of MHET and BHET [45], this is likely to pose a challenge for the biodegradation of PET using whole cells. Other solutions have been tested such as the use of mixtures of hydrolases that act synergistically [46], or the selective modification of amino acid residues of the polyester hydrolase involved in the interaction with the inhibitors [47]. These factors emphasise the need for obtaining enzymes, either by direct screening or by modification of existing ones, which are not susceptible to inhibition by MHET and BHET and can therefore be used to develop efficient bioprocesses using PET as the substrate.

### 2.2. Metabolism of TPA

TPA is transformed into PCA by the pathway encoded by the *tph* genes. These genes encode two sequential catabolic steps: the addition of two hydroxyl groups in positions 4 and 5 of TPA by the activity of the TPA dioxygenase TphA1A2A3 producing 1,6-dihydroxycyclohexa-2,4-diene-dicarboxylate (DCD), and the removal of the carboxyl group in position 6 by the action of the 1,2-dihydroxy-3,5-cyclohexadiene-1,4-dicarboxylate dehydrogenase TphB (Figure 2A). The genes responsible for those activities have been characterised in the actinomycete *Rhodococcus* sp. strain DK17 [48], in the β-proteobacteria *Comamonas testosteroni* YZW-D [49], and in *Comamonas* sp. strain E6 [50]. In addition to the catabolic *tph* genes, both organisms encode within this cluster the transcriptional regulator TphR (Figure 2B). TphR has been described as an IclR-type activator that responds to the inducer TPA [51]. *Comamonas* sp. strain E6 also contains the extra gene *tphC*, which encodes a permease involved in the uptake of TPA using the tripartite aromatic acid transporter [52].

We conducted a systematic analysis of the presence of the *tph* genes in the genomes available in public databases. As a result, we identified genes sharing a significant identity (greater than 35% for all the genes in the cluster with *tphA2* greater than 65%) and similar genetic organisation in only a limited number of organisms, which are representative of β-proteobacteria (*Comamonas*, *Ideonella* and *Ramlibacter*) and γ-proteobacteria (*Pseudomonas*), as well as of actinomycetes (*Rhodococcus*). In the genus *Rhodococcus* the *tph* genes are associated with plasmids with the exception of *Rhodococcus opacus* 1CP in which the cluster of genes was identified in the chromosome. In all the genomes investigated, the four catabolic genes were conserved in the same order. All clusters contain a regulatory gene encoding an IclR-type transcriptional regulator upstream the catabolic genes and in a divergent orientation. More diversity was observed in the putative transport of TPA inside the cell: all the β-proteobacteria utilized the transporter *tphC*, whereas the rest of the organisms contained a previously unidentified MFS transporter of the AAHS family (aromatic acid:H^+^ symporter; named *tphK*) homologous to the *p*-hydroxybenzoate transporter *pcaK* [53] (Figure 2B).

### 2.3. Metabolism of PCA

The PCA resulting from the activity of the Tph enzymes follows different pathways depending on the organism. This suggests that the *tph* genes can act as an independent metabolic module regardless of the type of PCA metabolism present in the TPA degrading strain. In fact, two copies of this cluster of genes are harboured by two different plasmids in *Rhodococcus* sp. strain DK17, indicating that this pathway can be mobilised by horizontal gene transfer into species containing one of the widespread PCA degradation pathways [48]. All PCA pathways share an initial dioxygenolytic step in which the aromatic ring is cleaved. Until now, three different pathways have been reported depending on the cleavage position in the aromatic ring. They are known as the *ortho*-, *meta*- and *para*-cleavage pathways and their initial reaction is catalysed by a PCA-3,4-, 4,5- and 2,3-dioxygenase, respectively (Figure 3) [27,29,54]. For simplicity, we will refer from now on to the nomenclature of the enzymes to discriminate between the pathways.

Using the sequences of characterised PCA dioxygenases, we conducted a bioinformatics search of the pathways likely involved in the metabolism of PCA that are present in the genomes in which we had previously identified the genes responsible for the conversion of TPA into PCA. Out of the three pathways, the PCA-2,3-dioxygenase was not present in any of them. Among the β-proteobacteria, *C. testosteroni*, *C. thiooxydans* and *R. tataouinensis* have homologs of the PCA-4,5-dioxygenase in their genomes, whereas *I. sakaiensis*, the different species of *Pseudomonas* and *R. opacus* contain the PCA-3,4-dioxygenase pathway. These results are consistent with previous observations showing that a PCA-3,4-dioxygenase activity is present in cells of *Rhodococcus* sp. strain DK17 growing on TPA [48], whereas a PCA-4,5-dioxygenase activity was identified in *Comamonas* sp. strain E6 [50]. Likewise, *I. sakaiensis* has been reported to contain a *tph* cluster and PCA-3,4-pathway [24].

The diversity of PCA metabolic pathways is an important factor when considering developing bioprocesses based on PET. Depending on the pathway used, a range of metabolites can be produced with different applications in mind. Out of them, the PCA-3,4-dioxygenolytic pathway has been thoroughly studied. This route is one of the branches of the β-ketoadipate pathway that connects the metabolism of aromatics converging on either catechol (e.g., benzoate) or PCA (e.g., 4-hydrozybenzoate) with the central metabolism of certain bacterial species [30]. The β-ketoadipate pathway has traditionally been used as a way of incorporating toxic and recalcitrant aromatic molecules in the central metabolism of bacteria, including nitrophenols and polychlorinated arenes. It is also an important path for funnelling the degradation products of lignocellulosic waste that could be used for the synthesis of other molecules of interest [56]. Strikingly, despite the metabolic diversity of the pathways involved which could allow for the production of molecules with interesting properties (e.g., functionalised lactones), complete mineralization of PCA continues to be the main application of the PCA metabolism. Only recently, PCA obtained from lignin-derived aromatics has been used for synthesis of the industrially relevant metabolite adipic acid [57]. This has not been achieved by the action of any of the described PCA pathways, but by the conversion of PCA into catechol catalysed by a PCA decarboxylase. Catechol is then transformed into *cis,cis*-muconate by the action of a catechol-1,2-dioxygenase, and the latter is hydrogenated abiotically to adipic acid in the presence of a catalyst [58].

### 2.4. Metabolism of EG

The metabolism of EG is more diverse compared to TPA. In acetogens, EG is oxidised to ethanol and acetaldehyde that is eventually converted to acetate via acetyl-CoA [31]. In other bacterial species, however, EG is degraded via the formation of glyoxylate (GLA) (Figure 4A) [59,60]. Activities responsible for the conversion of EG into GLA have been identified in multiple organisms. These initial steps are catalysed by dehydrogenases with broad specificity involved in the metabolism of short-chain alcohols and aldehydes such as the propanediol oxidoreductase of *E. coli* (also known as lactaldehyde reductase AldA) [61]. In *Pseudomonas aeruginosa* and *P. putida*, the initial reaction is carried out by periplasmic alcohol dehydrogenases that depend on pyrroloquinoline quinone for their activity [32,62]. Once GLA is produced, the pathway proceeds to intermediates of the central metabolism through different routes depending on the organism. For instance, whereas in *Escherichia coli* the pathway continues to acetyl-CoA via 3-phosphoglycerate—this is called the “canonical” pathway [63]—it has been proposed that some strains of *P. putida* make use of the shunt that funnels GLA to the Krebs cycle via isocitrate or malate [32,64]. The genetic determinants of the canonical GLA pathway have been identified in different microorganisms. The reactions are catalysed by the enzymes GLA carboligase (Gcl), tartronate semialdehyde reductase (GlxR) and glycerate-2-kinase (GlxK), all of which are encoded in the same cluster of genes in *E. coli* K12 and *Pseudonocardia dioxanivorans* strain CB1190 [65,66].

Using the sequences of FucO and Gcl from *E. coli* as probes, we conducted an analysis of the likelihood of the occurrence of activities for EG degradation in different bacteria. Homologs to *fucO* are widespread and present in all organisms investigated (not shown). Added to the broad substrate specificity of the enzymes active against EG, this suggests that EG degradation is a relatively common feature in bacteria. Likewise, the canonical pathway for GLA degradation seems ubiquitous as *gcl* is conserved in a very large number of bacterial species (not shown). As TPA degradation genes are not as frequently present in bacterial genomes, next we investigated the presence of activities for EG degradation in the strains that we had previously identified as carriers of the *tph* genes for TPA mineralisation (Figure 4B). All of them contain homologs to *fucO* or alcohol dehydrogenases similar to *pedE* described in *Pseudomonas* species. Moreover, all of them contain homologs to *gcl*, *glxR* and *glxK*, although only the genetic organization of these genes in *R. opacus* resembles that of *E. coli*. Contrary to the case of TPA, our synteny search did not identify conserved transporters involved in the uptake of EG or GLA. Likewise, no regulatory elements controlling the expression of the genes responsible for GLA degradation could be found.

Taken all together, these results indicate that most organisms capable of degrading TPA are also likely able to degrade EG, thereby enabling a more efficient usage of the products resulting from PET hydrolysis. In this sense, it has been recently demonstrated that EG can be readily transformed into the bioplastic polyhydroxyalkanoate in an engineered strain of *P. putida* KT2440 [64], underlining the usability of microorganisms for the conversion of oil-derived plastics into bioplastics.

## 3. Anabolism of Monomers Used for Bio-PET Synthesis

Bio-based PET, also known as bio-PET, is the common term used to refer to a PET polymer in which at least a fraction of the constituent monomers is obtained from biological—and therefore renewable—sources. In this section we will review recent efforts to produce TPA and EG involving microorganisms at any step (Figure 5). These methods can be fully or at least partially biotic and may involve abiotic physico-chemical steps. Even if not completely green, these synthetic processes promise to decrease the dependence on virgin PET derived from fossil feedstocks and may certainly contribute to a fully circular and sustainable PET economy.

### 3.1. Biosynthesis of TPA

The microbial biosynthesis of aromatic compounds has not been characterised with the same level of detail as their degradation. Despite this, there are a number of pathways that render aromatic compounds and generally involve the metabolism of aromatic amino acids and the shikimate pathway, or the condensation of molecules such as *cis,cis*-muconate [68]. Unfortunately, none of the currently known pathways are likely to allow for the direct production of TPA from central intermediates. It has been proposed, however, that the shikimate pathway could be used to produce *p*-toluate that could later be transformed into TPA, although the activities required for this pathway have not been identified [69]. Inspired by this, we have conducted a retrosynthesis analysis of plausible biochemical reactions that could render TPA using as substrates molecules present in the metabolism of *E. coli*. This allows for the formulations of reactions that are chemically plausible (e.g., because the mechanism involves reactive groups following known mechanistic rules), even though this might be in the absence of any biochemical evidence [70]. This method is particularly useful for guiding the screening of genomic or metagenomic libraries in search of genes coding for enzymes capable of catalysing a proposed reaction, as well as for the lab-directed evolution of known enzymes for the efficient catalysis of novel reactions. Our analysis resulted in a number of pathways leading to benzoate that can be obtained from phenylalanine, which itself is produced from shikimic acid [71]. The last step, however, will be more difficult to take place biotically as it would involve the conversion of benzoate into TPA by direct incorporation in the aromatic ring of a carboxylic group coming from bicarbonate, a step that is typically conducted at high temperatures and in the presence of metal catalysts [72,73].

Another possibility for the sustainable production of TPA is to use aromatics obtained from renewable sources such as lignin [74]. A recent work shows that TPA biosynthesis can be achieved from *p*-xylene [75]. This process was successfully implemented in *E. coli* by the heterologous expression of segments of two different pathways. In this process, *p*-xylene is first converted into toluic acid by the action of the xylene monooxygenase (XylMA), benzyl alcohol dehydrogenase (XylB) and the benzaldehyde dehydrogenase (XylC) of the TOL pathway for the degradation of toluene and xylene encoded in the pWW0 plasmid of *P. putida* mt-2 [76]. These enzymes oxidise, respectively, one of the methyl groups of xylene to a carboxylic group via formation of the corresponding alcohol and aldehyde [77]. Toluic acid is later transformed into TPA by the action of a toluene sulfate monooxygenase (TsaMB), a 4-carboxybenzaldehyde dehydrogenase (TsaC) and a 4-carboxybenzalcohol dehydrogenase (TsaD) present in *C. testosteroni* T2 [78].

This biosynthetic pathway poses a significant improvement in terms of sustainability compared to conventional chemical methods [75], but obtaining *p*-xylene from renewable sources also poses a considerable challenge. This has been solved by using isobutanol [79,80] or biomass as substrates for different chemical transformations. Pyrolysis of biomass [81], as well as the Diels-Alders condensation of ethylene with different types of biomass-derived molecules (e.g., furans) can be used to produce *p*-xylene or TPA [82,83,84,85]. Ethylene itself can be produced by different biosynthetic pathways, some of which have been harnessed to produce high levels of this molecule in engineered bacteria [86,87].

### 3.2. Biosynthesis of EG

Given the difficulties of obtaining TPA from sources other than fossil feedstocks, bio-PET typically refers to a PET polymer in which only EG is obtained from renewable sources [88]. EG accounts for 30% of the mass of the polymer and, therefore, this is usually the maximum percentage of bio components encountered in bio-PET. As recently reviewed in [89], there are a number of artificial pathways that have been engineered to obtain EG from renewable plant feedstocks using microorganisms. Among them, biosynthesis of EG in bacteria can be achieved in high yields by a pentose pathway that uses xylose as a substrate (Figure 5). Xylose is first transformed into xylonate by the action of a dehydrogenase. After the subsequent action of a dehydratase and an aldolase, glycoaldehyde is obtained, which is finally reduced to EG by a reductase [90,91,92]. This pathway has been extensively engineered to increase production yields that currently reach a 98% of the theoretical maximum and constitute a promising alternative for the synthesis of EG [93].

The engineered xylose pathway is not the only way of obtaining EG. It can also be produced from glucose in *Saccharomyces cerevisiae* using glycolytic enzymes [94] and via the synthesis of serine in an engineered pathway in *E. coli* [95]. Serine is transformed into ethanolamine by a plant serine decarboxylase. Ethanolamine is later transformed into glycolaldehyde by an oxidase and the latter reduced to EG by a reductase (Figure 5). The pathway has been artificially reconstituted in *E. coli* and is also amenable to metabolic engineering efforts to improve production yields. More recent work has shown the feasibility of using synthesis gas (syngas) for the production of EG harnessing the Wood-Ljungdahl pathway of carbon fixation present in acetogenic bacterial species such as *Moorella thermoacetica* and *Clostridium ljungdahlii* [96]. In another approach, EG was obtained from gaseous alkenes by a strain of *E. coli* that expresses recombinantly a monooxygenase and an epoxide hydrolase [97].

Similarly to TPA, EG can also be directly obtained from biomass. This can be achieved through the dehydration of cellulosic ethanol [98], the hydrogenolysis of xylitol [99] and the hydrogenation of corn stalk [100]. This reflects a wide diversity of options for the production of EG that could be used to replace the chemical procedures relying on fossil feedstocks.

## 4. Future Prospects and Concluding Remarks

Here we have reviewed the potential use of PET as a feedstock for microbial biotransformations. We have identified the challenges of large-scale PET enzymatic hydrolysis and proposed strategies for the enhancement of this process using enzymes—and possibly organisms—capable of being stable and active near the Tg of the polymer.

Similarly to the case of glycolytic procedures, TPA and EG resulting from hydrolysis could be used for the synthesis of fresh PET, but we also advocate for their biotransformation into molecules or processes with added value. An example of this could be their use in microbial fuel cells for the production of electricity that has been achieved using TPA as a carbon source [101]. TPA metabolism is neither widespread nor diverse in the genomes currently available. This could constitute a bottleneck for the development of future applications that currently have adipic acid as the main target of molecules funnelled through the PCA pathways. EG metabolic genes, on the contrary, are found in numerous organisms and encode a more diverse metabolism, likely enabling a variety of applications.

Hand in hand with an efficient degradation of PET, a circular economy of this polymer requires a sustainable large-scale synthesis of TPA and EG. We have reviewed a number of efforts made for the biosynthesis of bio-PET using renewable sources. On this front a milestone seems to have been reached recently with the production of the first bottle that is completely made of monomers obtained from biological sources [102]. Any method, including those reaching a maximum of 30% bio-PET, have a significantly lower carbon footprint compared with the synthesis of oil-derived plastics and are worth pursuing. By using plant biomass, it is possible to contribute to CO_2_ fixation, although a major breakthrough would be to obtain TPA or EG with engineered microorganisms directly from CO_2_.

Overall, the prospects for a circular bio-based economy of PET are encouraging, and most of the technological hurdles for either biodegradation or biosynthesis have already been overcome, or there are alternatives or clear strategies to overcome them. Although bio-approaches to the PET economy might not be as profitable as the current *status quo* in the short term, there is an undeniable pressure from the general public to manage PET differently, and this is already producing changes in policies and regulations. In our view, this will at least partially bridge the strict financial gap compared to chemical processes, which will enable itself the diversification of applications of PET including its upcycling in other molecules. In the long run, this will have a positive impact on recycling rates and will also lower the environmental release of PET waste, therefore contributing to solving an imperative environmental concern. 

## Figures and Tables

**Figure 1 genes-10-00373-f001:**
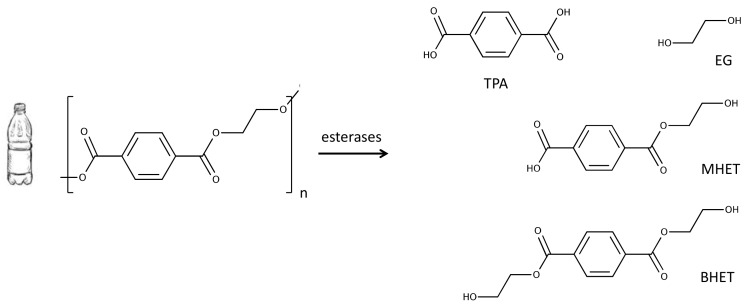
Enzymatic hydrolysis of polyethylene terephthalate (PET) results in a mixture of terephthalic acid (TPA) and ethylene glycol (EG) and, to a lesser extent, the incomplete hydrolysis products bis-(2-hydroxyethyl) terephthalate (BHET) and mono-(2-hydroxyethyl) terephthalate (MHET).

**Figure 2 genes-10-00373-f002:**
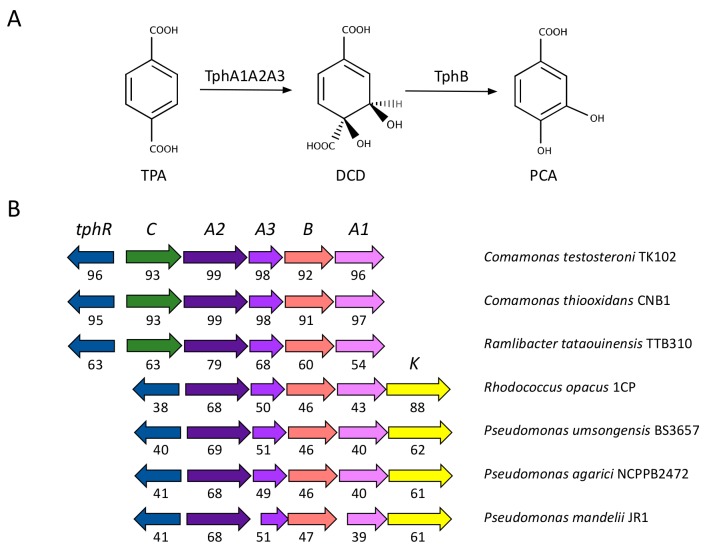
(**a**) TPA metabolism reported in bacteria. The names of the molecules and abbreviations are: terephthalic acid, TPA; 1,6-dihydroxycyclohexa-2,4-diene-dicarboxylate, DCD; protocatechuate, PCA. (**b**) Genetic organisation of the *tph* genes identified in several genomes available in databases. Numbers below arrows indicate the percentage of identity compared to the orthologous genes present in *Comamonas* sp. E6 (accession: AB238679; [50]) with the exception of the *tphK* genes that were compared to the ortholog present in the plasmid pDK3 of *Rhodococcus* sp. DK17 (accession: AY502076; [48]). Plots were produced with SyntTax (http://archaea.u-psud.fr/SyntTax; [55]).

**Figure 3 genes-10-00373-f003:**
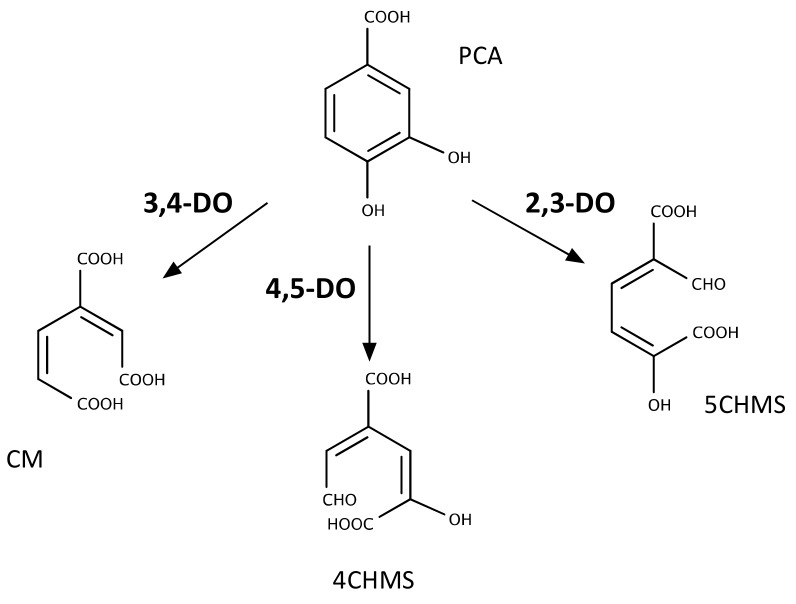
Types of dioxygenase-mediated reactions involved in PCA cleavage by bacteria. DO: dioxygenase; CM: 3-carboxy-cis,cis-muconate; 4CHMS: 4-carboxy-2-hydroxymuconate semialdehyde; 5CHMS: 5-carboxy-2-hydroxymuconate-6-semialdehyde.

**Figure 4 genes-10-00373-f004:**
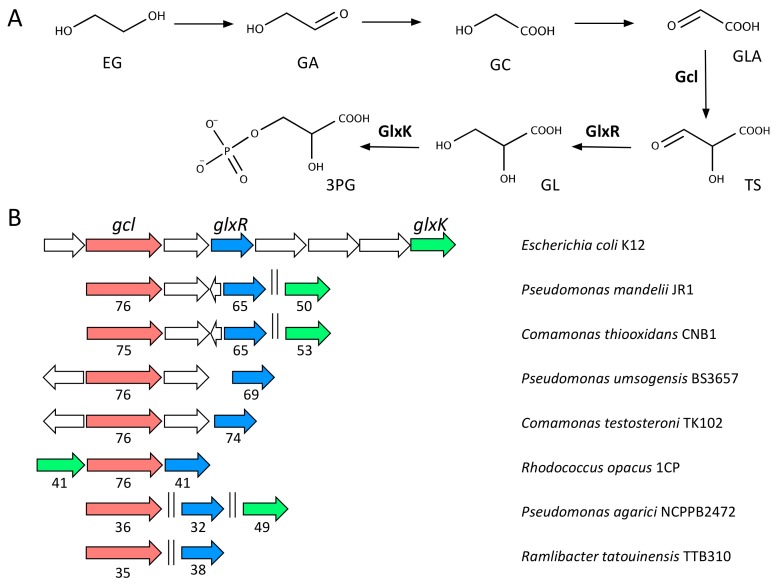
(**a**) EG metabolism via GLA. The GLA canonical pathway described in the text is shown. The 3-phosphoglycerate (3PG) produced is later funnelled into the central metabolism via acetyl-CoA. The names of the molecules and abbreviations are: ethylene glycol, EG; glycoladehyde, GA; glycolate, GC; glyoxylate, GLA; tartronate semialdehyde, TS; glycerate, GL. (**b**) Genetic organisation of the genes involved in GLA metabolism identified in several genomes available in databases. Numbers below arrows indicate the percentage of identity compared to the orthologous genes present in *E. coli* K12 (accession: AP009048; [67]). Plots were produced with SyntTax (http://archaea.u-psud.fr/SyntTax; [55]).

**Figure 5 genes-10-00373-f005:**
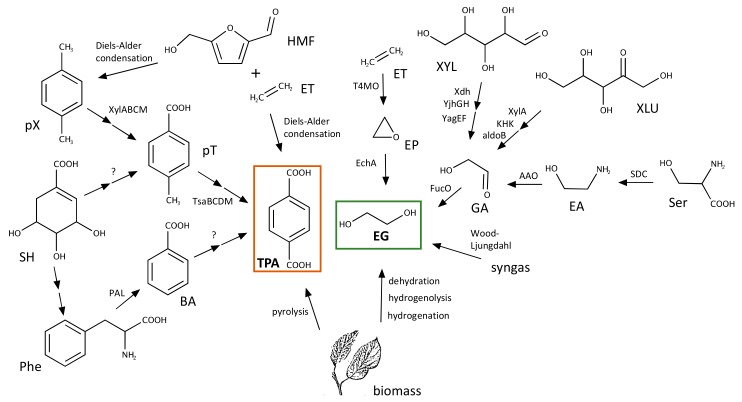
Selected pathways and processes used to produce TPA and EG from renewable sources. The names of the molecules and abbreviations are: *p*-xylene, pX; *p*-toluate, pT; shikimate, SH; phenylalanine, Phe; benzoate, BA; 5-(hydromethyl)furfural, HMF; ethylene, ET; ethylene oxide, EP; xylose, XYL; xylulose, XLU; glycoaldehyde, GA; serine, Ser; ethanolamine, EA. If known, the names of enzymes/processes responsible for the different conversions are shown next to the arrows.

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
