# Peer review of "Microbial Genes for a Circular and Sustainable Bio-PET Economy"

_genes, 2019, doi:10.3390/genes10050373_

Round 1

Reviewer 1 Report

This is a really nicely and clearly written review on PET degradation/synthesis. Each one of the steps of PET biodegradation is excellently described. While PET biosynthesis remains challenging, it has been nicely acknowledged by the authors (as well as many other aspects regarding the field of plastic degradation e.g. crystallinity, etc). The structure of the text and information given is concise with all relevant and up to date references. Authors are clearly knowledgeable in this topic and this is clearly reflected. While I am usually a meticulous reviewer who always has at least some minor comments when reviewing manuscripts, I have nothing to suggest to such excellent piece of work. I can only congratulate the authors!

Author Response

We do appreciate the comments made by the reviewer and we are pleased to learn that the reviewer found it interesting. We are extremely grateful for the time and effort they have spent reviewing this manuscript.

Reviewer 2 Report

Manuscript  genes-501635 is an interesting review, concise and well-written.

I have only minor aspects to comment or suggest:

-          The tittle is a bit confusing, I would suggest to change “Genes for a circular…” for “Microbial genes for a circular…” , it is more appropriate for the topic of this review.

-          Line 134, specify what temperatures

-          Line 137, specify the temperature range of these “mesophilic conditions”

-          Lines 165-166,   include the degree of identity considered significant in this study.

-          Figure 1 legend, explain the abbreviations BHET and MHET .

Author Response

We thank the reviewer for the positive comments and the time and effort invested in conducted this review.

We have incorporated the feedback as follows:

- We have included the word 'microbial' at the beginning of the title as per the reviewer suggestion.

- The glass transition range of temperatures of PET has been included in line 131 and we have added the following sentence for clarity in line 134: 'Both of which, specially the latter, have been reported to be active for long periods of time at temperatures of up to 70°C'.

- The mesophilic conditions of growth have been modified to include the following sentence and references 'which can grow at up to 48-50°C only after evolutionary adaptation and at a fitness cost [40,41]'

- A cluster of genes was considered for comparison when all its genes had an identity greater that 30% and the identity of tphA2 was greater than 65%. These values have been included in line 215.

- Full names of BHET and MHET have been included in the legend of Fig. 1.